# Topical Melatonin Exerts Immunomodulatory Effect and Improves Dermatitis Severity in a Mouse Model of Atopic Dermatitis

**DOI:** 10.3390/ijms23031373

**Published:** 2022-01-25

**Authors:** Yung-Sen Chang, Chih-Chen Tsai, Pang-Yan Yang, Chih-Yu Tang, Bor-Luen Chiang

**Affiliations:** 1Department of Pediatrics, Taipei City Hospital Renai Branch, Taipei 106, Taiwan; yschang1018@gmail.com (Y.-S.C.); davis456258@gmail.com (C.-C.T.); stevenno22@hotmail.com (P.-Y.Y.); bill90810@gmail.com (C.-Y.T.); 2Department of Pediatrics, National Taiwan University Hospital, Taipei 100, Taiwan; 3School of Medicine, National Yang Ming Chiao Tung University, Taipei 112, Taiwan; 4Department of Psychology and Counseling, University of Taipei, Taipei 100, Taiwan; 5Graduate Institute of Clinical Medicine, College of Medicine, National Taiwan University, Taipei 100, Taiwan

**Keywords:** atopic dermatitis, melatonin, IP-10, DNCB-induced dermatitis, HaCaT cells

## Abstract

Oral melatonin supplement has been shown to improve dermatitis severity in children with AD, but the mechanism of the effect is unclear, and it is uncertain whether melatonin has a direct immunomodulatory effect on the dermatitis. Topical melatonin treatment was applied to DNCB-stimulated Balb/c mice, and gross and pathological skin findings, serum IgE, and cytokine levels in superficial lymph nodes were analyzed. Secretion of chemokines and cell proliferative response after melatonin treatment in human keratinocyte HaCaT cells were also studied. We found that in DNCB-stimulated Balb/c mice, topical melatonin treatment improved gross dermatitis severity, reduced epidermal hyperplasia and lymphocyte infiltration in the skin, and decreased IP-10, CCL27, IL-4, and IL-17 levels in superficial skin-draining lymph nodes. Melatonin also reduced cytokine-induced secretion of AD-related chemokines IP-10 and MCP-1 and decreased IL-4-induced cell proliferation in HaCaT cells. Melatonin seems to have an immunomodulatory effect on AD, with IP-10 as a possible target, and topical melatonin treatment is a potentially useful treatment for patients with AD.

## 1. Introduction

Atopic dermatitis (AD), also called eczema, is a chronically relapsing pruritic inflammatory skin disease affecting 15–30% of children and 2–10% of adults, and its prevalence is continuously increasing [1]. Disturbed sleep is reported in 47–60% of AD patients and is a major factor leading to an impaired quality of life [2,3,4]. Our previous studies have found that children with AD had significantly reduced sleep efficiency, longer sleep onset latency, more sleep fragmentation, and less nonrapid eye movement sleep [5]. We also found that in AD patients, a lower nocturnal melatonin level was associated with poor sleep efficiency, more sleep fragmentation, and more severe skin disease [5]. Melatonin is a hormone secreted by the pineal gland that is essential for regulating sleep [6]. Oral melatonin has a sedative effect and has been used for the management of insomnia and jet lag. Previous studies have shown that melatonin could shorten sleep onset latency and increase total sleep time and sleep efficiency [7,8,9,10,11,12]. Melatonin also has immunomodulatory, anti-inflammatory, and anti-oxidative effects [13,14,15]. which might improve the skin inflammation and help maintain a functional epidermal barrier in AD patients [15,16,17]. Therefore, our team conducted a randomized, double-blind, placebo-controlled crossover study to investigate whether melatonin is effective for improving skin inflammation and sleep in children with atopic dermatitis. We found that oral melatonin supplement significantly improved AD disease severity, on average lowering the disease severity score, the Scoring Atopic Dermatitis (SCORAD) index by 9.9 in 4 weeks compared with placebo (95%CI, −13.7 to −4.6; *p* < 0.001). Melatonin supplement also significantly shortened the sleep onset latency by 23.4 min compared with placebo (95%CI, −38.6 to −4.2; *p* = 0.02) [18]. We have shown that the use of melatonin is potentially a very promising new treatment strategy in children with AD, as it improves both skin inflammation and sleep. However, the mechanism of how melatonin improves AD and whether melatonin has a direct effect on the inflammation of the skin in AD patients remain unknown and needs to be clarified. In this study, we further investigated the effect of topical melatonin on the alleviation of allergic skin inflammation in a murine model of atopic dermatitis.

## 2. Results

### 2.1. Animal Study

Balb/c mice stimulated with DNCB developed AD-like skin lesions with erythema, edema, excoriation, and dryness. (Figure 1a) In mice treated with melatonin, the gross dermatitis clinical score was significantly reduced (Figure 1b), and analysis of separate scoring showed that the edema and dryness scores significantly improved after melatonin treatment. (Figure 1c). Ear thickness was also significantly decreased after melatonin treatment. (Figure 1d) Pathological results found that DNCB-stimulated skin had severe epidermal hyperplasia and marked infiltration of lymphocytes and mast cells. These inflammatory findings were less prominent in those treated with melatonin. (Figure 2).

Serum total IgE levels were significantly increased in mice stimulated with DNCB, and those treated with melatonin had mildly decreased levels of serum IgE, but this was not statistically significant. (Figure 3) Levels of IP-10, CCL-27, IL-4 in the superficial inguinal lymph nodes were significantly increased in mice stimulated with DNCB, and those treated with melatonin had significantly reduced levels of these AD-related cytokines and chemokines. IL-17 level in the superficial inguinal lymph nodes were also significantly decreased in those treated with melatonin (Figure 4).

### 2.2. Cell Experiment

To determine the non-cytotoxic concentration of melatonin on HaCaT keratinocytes, we used the MTT assay. Results showed that the cell viability of HaCaT cells was above 90% when treated with 2 mM melatonin for 12 h. (Figure 5) Melatonin treatment significantly decreased TNF-α and IFN-γ induced MCP-1 production in HaCaT cells in a dose-dependent manner. Melatonin treatment also significantly decreased IL-4, IL-13, and IFN-γ induced IP-10 secretion in a dose-dependent manner but did not significantly decrease IL-17 secretion. (Figure 6) Melatonin treatment also significantly reduced IL-4 induced proliferation of HaCaT cells. (Figure 7).

## 3. Discussion

In a previous randomized, double-blind, placebo-controlled study, our team found that AD disease severity and sleep onset latency improved after oral melatonin treatment in children with AD [18]. However, total sleep time, sleep efficiency, and scratching activity during sleep were not significantly improved [18]. Therefore, it is likely that the improvement in AD disease severity is not solely due to the effect of melatonin on the improvement in sleep, and could be due to its immunomodulatory effects [16,17].

The pathogenesis of AD is complex and involves a variety of immune cells, cytokines and chemokines [16,17]. In acute lesions, Th2 and Th22 responses are dominant, and IL-4 and IL-13 play major roles. In chronic lesions, there is also activation of the Th1 response and elevated IFN-γ, CXCL9, and IP-10 levels. In both acute and chronic AD, Th17-associated response is up-regulated, and IL-17A, PI3/elafin, and CCL20 are elevated [19]. Serum IL-31, CCL17, CCL22, and CCL27 levels have been found to be correlated with AD disease activity [19]. Histamine, IL-33, IL-31, IL-4, IL-13, and thymic stromal lymphopoietin (TSLP) have been suggested to be mediators of pruritus in AD [20]. MCP-1 is expressed in keratinocytes in patients with AD and plays an important role in monocyte and macrophage trafficking [21]. Various immunomodulatory mechanisms of melatonin have been reported [16,17]. Both Th1 and Th2 cells express membrane and nuclear receptors for melatonin; via these receptors, melatonin induces the synthesis of IFN-γ, IL-2, IL-6, and IL-12 by lymphocytic and monocytic cell lines and amplifies melatonin receptors. [15,22,23]. Melatonin also influences the activity of NK cells, T and B lymphocytes, granulocytes, monocytes, and mast cells [15,24]. Furthermore, through the upregulation of antioxidant enzymes, melatonin efficiently neutralizes several free radicals and stabilizes cell membranes [25]. However, how melatonin might modulate the complex inflammatory pathways in atopic dermatitis has not been extensively explored. A study has showed that melatonin suppresses the development of AD-like dermatitis in DNFB-treated NC/Nga mice, reduces total serum IgE level, and decreases IL-4 and IFN-γ production by activated CD4+ T cells [26]. In our study, we found that in Balb/c mice with AD-like skin lesions induced by DNCB, melatonin treatment improved the gross dermatitis severity, reduced lymphocyte infiltrations in the skin, and the levels of AD-related cytokines and chemokines such as IP-10, CCL-27, IL-4, and IL-17 were significantly decreased in the superficial skin-draining lymph nodes after melatonin treatment. From our cell experiments with the human keratinocyte cell line HaCaT cells, we also found that melatonin treatment could reduce the cytokine-stimulated secretion of AD-related cytokines, including TNF-α and IFN-γ induced MCP-1 production, and IL-4, IL-13, and IFN-γ induced IP-10 secretion. We have shown that melatonin could exert immunomodulatory effects on AD which might improve the dermatitis.

IP-10, also known as CXCL10, is a chemokine which could bind to the CXCR3 receptor to induce chemotaxis, cell growth, apoptosis, and angiostasis [27]. IP-10 has been found to be elevated in both the plasma and the interstitial fluid in lesional AD skin [28,29]. A study also found that IP-10 level in lesional AD skin was significantly correlated with the severity scoring index SCORAD [29]. Drugs used for treating AD have been shown to modulate IP-10: Previous studies have shown that antihistamines inhibited the production of IP-10 in human monocyte-derived dendritic cells and autologous CD4+ T cells, [30]. and tacrolimus has been found to suppress LPS-induced IP-10 in monocytes [31]. In our previous randomized controlled trial [18]. we found that serum IP-10 levels were significantly decreased after oral melatonin supplement in children with AD (unpublished data). In our present study, IP-10 is also a consistent target after melatonin treatment in both our animal and cell experiments. Superficial inguinal lymph node IP-10 level significantly decreased after melatonin treatment in DNCB-stimulated Balb/c mice with AD-like lesions; and IL-4, IL-13, and IFN-γ induced IP-10 secretion was also suppressed in HaCaT cells treated with melatonin. IP-10 is a potential target which might play a role in the mechanism of the effect of melatonin on AD.

Atopic dermatitis is characterized by scaly, cracked, and thickened skin. In our study, DNCB-stimulated mice also had thickened skin which improved after melatonin treatment. Ear thickness was significantly reduced, and the pathological examination showed that epidermal hyperplasia and lymphocyte infiltration of the epidermis were decreased. In our cells experiments, melatonin significantly reduced cell proliferation induced by IL-4 in HaCaT cells. Therefore, the improvement in skin thickening after melatonin treatment in AD could be contributed to the combined effect of reduced keratinocyte proliferation, and regulation of chemotactic chemokines such as IP-10 and MCP-1.

Currently the mainstay of treatment for AD is topical steroids. However, there is often poor compliance in children since the parents are often over-concerned about the side effects of steroids. The skin-thinning side effect of potent topical steroids also leads to limitations of its use in delicate areas such as the face, eyelids, and groin. Topical calcineurin inhibitors are used as a second-line treatment; however, there are a wide range of side effects and there have been concerns about possible carcinogenicity [32]. There is need for alternative treatment options for the treatment of children with AD. Our previous study has shown that oral melatonin improved disease severity in children with AD [18], but topical melatonin treatment has not been tried in patients with AD. Previous studies on topical melatonin showed that it had protective effect against the UVR-induced skin damage including DNA repair [33], promoted ulcer healing in patients with gastroesophageal reflux disease and peptic ulcer disease [34] and improved androgenetic alopecia [35,36]. From our current study, topical melatonin treatment improved the skin inflammation and thickening in a mouse model of AD, which is potentially due to regulation of inflammatory cytokines and chemokines including IP-10, MCP-1, CCL-27, and IL-4, and reduction of cytokine-induced cell proliferation. Topical melatonin has great potential to become a novel treatment method for AD patients. Further study is needed to evaluate whether it is clinically effective.

## 4. Materials and Methods

### 4.1. Animal Study

#### 4.1.1. Experimental Mice

Four to six-week-old Balb/c mice were maintained under specific pathogen-free conditions. Animals were housed in an air-conditioned animal room (25 ± 1 °C; relative humidity 40 ± 5%) and were fed a laboratory diet and distilled water. All the animal experiments were approved by the Institutional Animal Care and Use Committee of National Taiwan University College of Medicine and College of Public Health, and all procedures were conducted in accordance with the U.S. National Institute of Health guidelines.

#### 4.1.2. Materials

All materials used were of the analytical grade commercially available. 2,4-Dintrochlorobenzene (DNCB) and melatonin were purchased from Sigma-Aldrich Co. (St. Louis, MO, USA) and dissolved in acetone-olive oil (4:1 *v*/*v*).

#### 4.1.3. Induction of AD-like Skin Lesions

The procedure for induction of dermatitis is modified from the protocol previously described [37]. The backs of mice were shaved with an electric clipper and depilatory cream a day before DNCB sensitization. Dermatitis was induced by application of 20 μL of 1% DNCB to the outer and inner surfaces of mouse ears and 100 μL of the same solution to the shaved back skin once on days 1, 2, and 3 for sensitization. Sensitized mice were then challenged by applying 0.2% DNCB to the same skin surfaces daily from day 8 to day 20. The mice were sacrificed on day 21 for evaluation.

#### 4.1.4. Topical Melatonin Treatment

Mice were randomly assigned to the treatment and the control group. Each group consisted of 5 mice. For the treatment group, 20 μL of 10% melatonin was applied to the outer and inner surfaces of mouse ears and 100 μL of the same solution to the shaved back skin twice daily from day 8 to day 20. For the control group, the solution acetone-olive oil (4:1 *v*/*v*) was applied.

#### 4.1.5. Dermatitis Severity Evaluation

Skin lesion severity was evaluated daily. On day 21, ear thickness was measured with a thickness gauge (Digimatic Indicator, Mitsutoyo, Tokyo, Japan). Macroscopic assessment of the severity of dermatitis was determined by a clinical scoring method previously described [26]. The degree of erythema (hemorrhage), edema, excoriation (erosion), and dryness (scaling) were each scored as 0 (none), 1 (mild), 2 (moderate), and 3 (severe). The total score (0 to 12) was recorded. Assessment was performed by an investigator who is blind to the grouping of the animals.

#### 4.1.6. Histopathological Analysis

The treated skin and ears of mice were removed and fixed in 10% neutral-buffered formalin and embedded in paraffin. Then, 4-µm-thick sections were cut and transferred onto slides. Deparaffinized skin sections were stained with hematoxylin and eosin (H&E) before they were examined at 100× magnification for histopathological analysis.

#### 4.1.7. Measurement of Serum and Tissue Immunoglobulins

Blood samples were collected from the orbital sinus on Day 21 and stored at −80 °C after centrifugation. The serum immunoglobulin E (IgE) levels were measured using the ImmunoCAP fluorescence enzyme immunoassay (Phadia, Sweden) according to the manufacturer’s instructions. The superficial skin-draining axillary and inguinal lymph nodes of the mice were collected, and the levels of IFN-γ-inducible protein 10 (IP-10), CCL-27, interleukin (IL)-4, and IL-17 of the tissue were measured by enzyme-linked immunosorbent assay (ELISA) kits (R&D Systems, Minneapolis, MN, USA). Briefly, the tissue was homogenized in lysis buffer, and then the freezing/thawing procedure was repeated three times. After centrifugation, the supernatants containing total cellular protein was quantified and used to detect the level of chemokine and cytokines. Results were normalized to the total amount of protein prepared from tissue lysates.

### 4.2. Cell Experiment

#### 4.2.1. Cell Culture

HaCaT cells, a human keratinocyte cell line, were cultured in Dulbecco’s Modified Eagle Medium (DMEM) supplemented with glucose, l-glutamine, pyridoxine hydrochloride (Gibco, Waltham, MA, USA), 10% fetal bovine serum (Gibco, Waltham, MA, USA) and 1% penicillin/streptomycin/amphotericin antibiotic solution. The cells were maintained at 37 °C in a humidified atmosphere with 5% CO_2_.

#### 4.2.2. Cell Viability Assay

Cell viability at different concentrations of melatonin were determined using MTT assays. Melatonin was dissolved in dimethyl sulfoxide (DMSO) to 100 mM and then diluted with cell medium to the target concentration. Cells (1 × 10^4^/well) were seeded in triplicate into 96-well culture plates. After 24 h, cells were incubated at 37 °C in the presence or absence of melatonin at different concentrations (1–10 mM). Twelve hours later, 0.5 mg/mL of MTT solution was added to each well. After 3 h of incubation at 37 °C, supernatants were removed and 0.1 mL DMSO was added and shaken at low speed until crystals dissolved completely. Optical density was measured at a wavelength of 570 nm using an ELISA reader.

#### 4.2.3. Cytokine and Chemokine Response to Melatonin

HaCaT cells were plated in a volume of 2 mL medium in 6-well plates (1 × 10^6^/well) and stimulated with interferon-γ (IFN-γ) (10 ng/mL) and tumor necrosis factor-α (TNF-α) (10 ng/mL) or IFN-γ(100 ng/mL), IL-4(100 ng/mL), and IL-13(100 ng/mL). Three hours later, melatonin in different concentrations (0.5, 1, and 2-mM) was added in each well. Twelve hours later, cell supernatants were collected and the concentrations of monocyte chemoattractant protein-1 (MCP-1), IP-10, and IL-17 were measured by ELISA (R&D Systems) according to the standard protocols. Optical densities were measured at 450 nm. Each sample was tested in triplicate.

#### 4.2.4. Cell Proliferation Response to Melatonin

HaCaT cells were seeded in 96-well culture plates (1 × 10^4^/well) and stimulated with IL-4 (100 ng/mL). The medium was refreshed with IL-4 (100 ng/mL) at the 48th hour. Supernatants were removed at the 60th hour and IL-4 (100 ng/mL) and different concentrations of melatonin (0.5, 1, and 2-mM) were added. Twelve hours later, samples were collected, and the BrdU cell proliferation assay (Roche) was performed according to the manufacturer’s instructions. Optical densities were measured at 450 nm by the ELISA reader. Each sample was tested in triplicate.

#### 4.2.5. Statistical Analysis

Average data were presented as mean ± SD. Statistical analysis was completed using IBM SPSS for MAC, version 20. Comparisons between different experimental groups were conducted using the Student’s t test. Statistical difference is considered significant when *p* < 0.05.

## 5. Conclusions

In a mouse model of AD, topical melatonin treatment improved gross dermatitis severity, reduced epidermal hyperplasia and lymphocyte infiltration in the skin, and decreased IP-10, CCL27, IL-4, and IL-17 levels in superficial skin-draining lymph nodes. Melatonin also reduced cytokine-induced secretion of AD-related chemokines IP-10 and MCP-1, and decreased IL-4-induced cell proliferation in HaCaT cells. Melatonin seems to have an immunomodulatory effect on AD, with IP-10 as a possible target, and topical melatonin treatment is potentially useful for patients with AD.

## Figures and Tables

**Figure 1 ijms-23-01373-f001:**
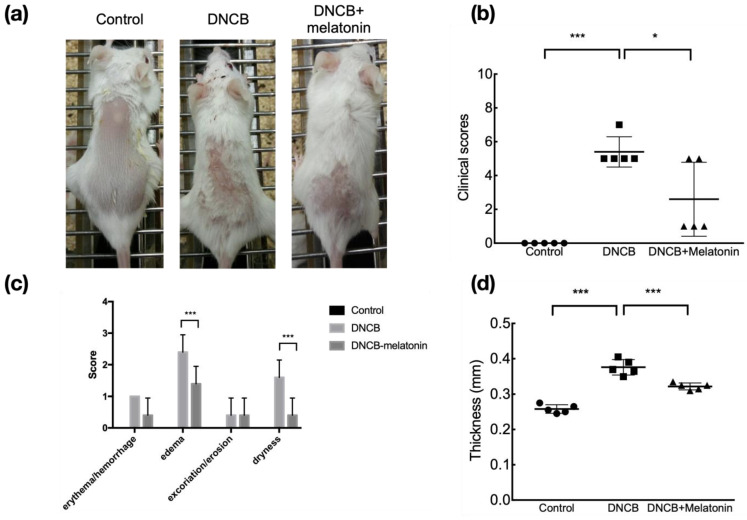
Gross skin lesions and clinical score of DNCB-stimulated mice with and without melatonin treatment. Mice stimulated with DNCB developed AD-like skin lesions, and those treated with melatonin had less severe dermatitis (**a**), significantly lower the clinical score (**b**), and significantly reduced ear thickness (**d**). Analysis of separate clinical scores showed that edema and dryness scores significantly reduced after melatonin treatment (**c**). * *p* < 0.05; *** *p* < 0.001.

**Figure 2 ijms-23-01373-f002:**
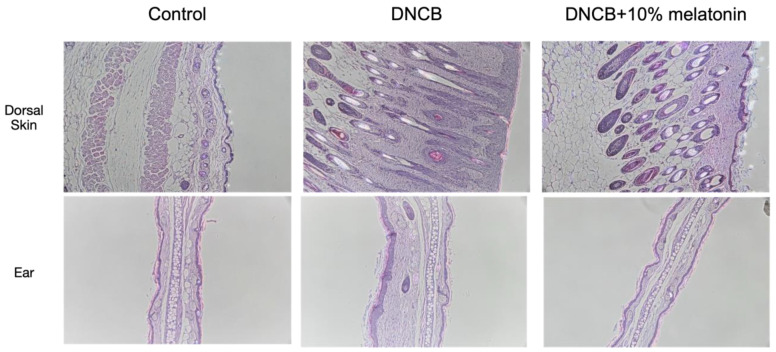
Pathology findings of the dorsal skin and ears of DNCB-stimulated mice with and without melatonin treatment. DNCB-stimulated mice had severe epidermal hyperplasia and marked infiltration of lymphocytes and mast cells in the skin, and increased ear thickness. Inflammation and thickening were less prominent in those treated with melatonin.

**Figure 3 ijms-23-01373-f003:**
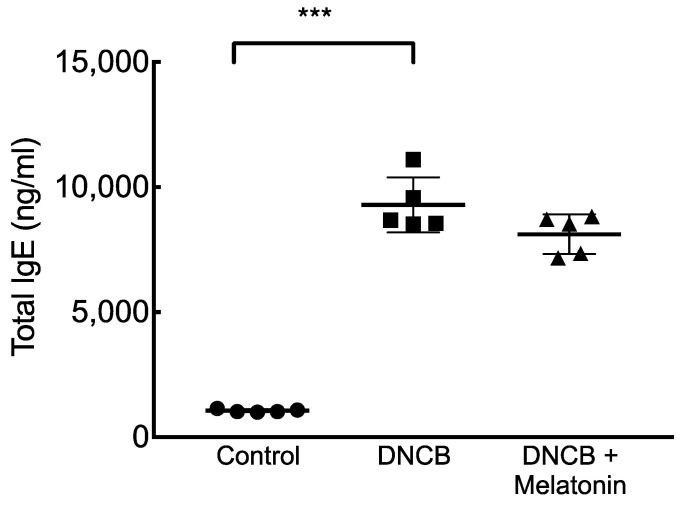
Serum total IgE levels in DNCB-stimulated mice with and without melatonin treatment. Serum total IgE levels increased in DNCB-stimulated mice, but only mildly reduced in those treated with melatonin. *** *p* < 0.001.

**Figure 4 ijms-23-01373-f004:**
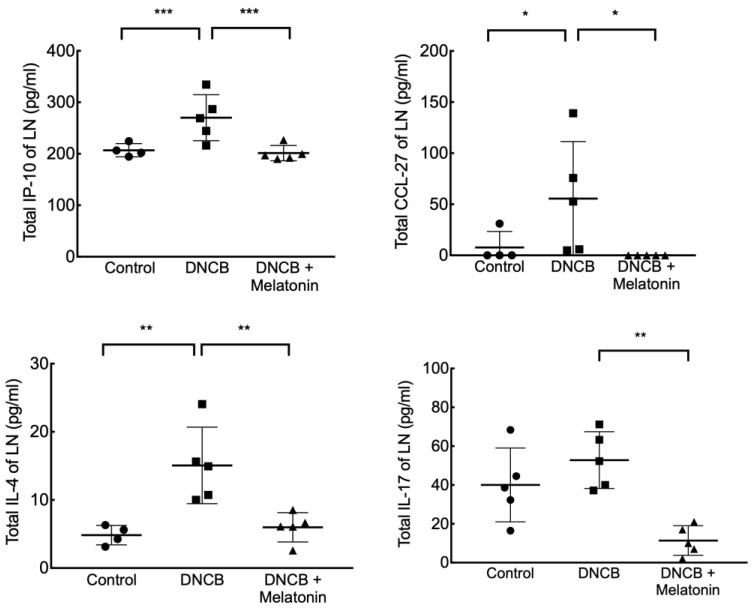
Inguinal lymph node levels of IP-10, CCL27, IL-4, and IL-17 in DNCB-stimulated mice with and without melatonin treatment. Melatonin treatment significantly decreased the DNCB-stimulated elevation of IP-10, CCL27, and IL-4 in inguinal lymph nodes, and significantly reduced IL-17 levels in DNCB-stimulated mice. * *p* < 0.05; ** *p* < 0.01; *** *p* < 0.001.

**Figure 5 ijms-23-01373-f005:**
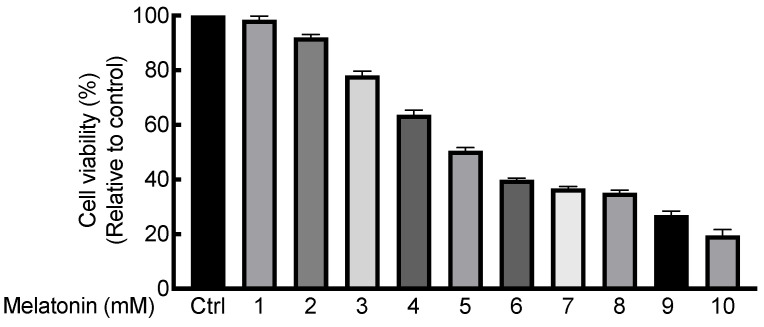
Cell viability of HaCaT cells under various concentrations of melatonin treatment for 12 h. Cell viability was kept above 90% when treated with 2 mM melatonin for 12 h.

**Figure 6 ijms-23-01373-f006:**
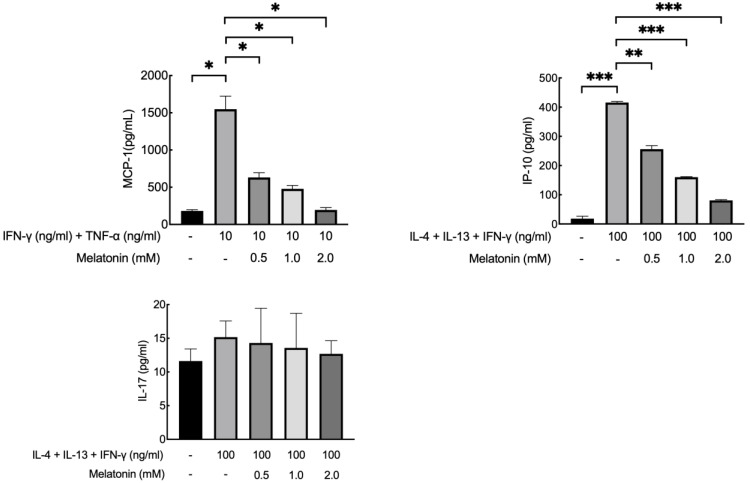
Effect of melatonin on cytokine induced MCP-1, IP-10, and IL-17 levels in HaCaT cells. Melatonin treatment significantly decreased MCP-1 and IP-10 production in stimulated HaCaT cells but did not significantly decrease IL-17 levels. * *p* < 0.05; ** *p* < 0.01; *** *p* < 0.001.

**Figure 7 ijms-23-01373-f007:**
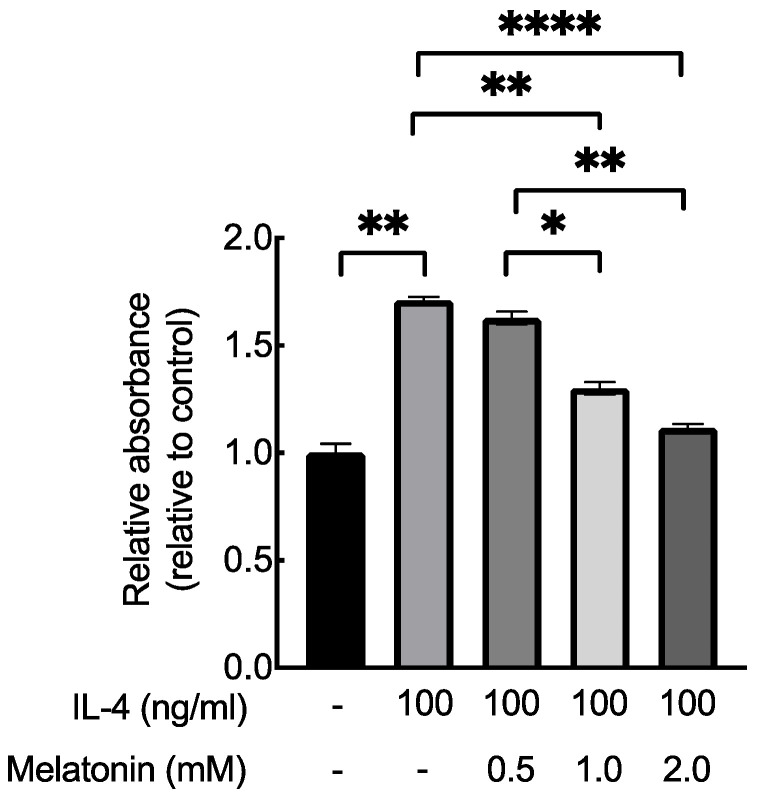
Effect of melatonin on IL-4 induced cell proliferation in HaCaT cells. IL-4 stimulated cell proliferation in HaCaT cells, and melatonin significantly reduced the cell proliferation in a dose-dependent manner. * *p* < 0.05; ** *p* < 0.01; **** *p* < 0.0001.

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
