# Peer review of "Topical Melatonin Exerts Immunomodulatory Effect and Improves Dermatitis Severity in a Mouse Model of Atopic Dermatitis"

_ijms, 2022, doi:10.3390/ijms23031373_

Round 1

Reviewer 1 Report

Very well-conducted original study and clear presentation in the original article. Introduction covered the significance of conducting this study, which builds on the previous work completed by this group. Methodology and statement of ethics application and origin of study resources were described clearly. The presentation of results was easy to follow. The discussion was very thorough and covered the physiological aspects of this topic very well. The authors highlighted current practice at the end to manage this condition, and summarized the potential how melatonin can be added as another treatment for this - though recognize further studies of its clinical application is required. 

No issues with language

Author Response

Thank you so much for your encouraging comments. From our study we found that topical melatonin is potentially useful for the treatment of atopic dermatitis, and we are preparing for further clinical studies addressing this issue.

Reviewer 2 Report

The tryptophan-derived hormone melatonin is well known as an important regulator of sleep-wake cycles and as such is used for treatment of some sleep disorders and to reduce jet lag. Melatonin was also reported as a potent antioxidant and free radical scavenger. There is increasing evidence that melatonin interacts with the immune system and has some immunomodulary activity. However, these interactions have been scarcely explored so far.

In this study presented by Chang et al. the effects of melatonin were investigated in human keratinocyte cell line (HaCaT cells) in vitro and in a mouse model of atopic dermatitis (AD) in vivo.

The authors observed that topical melatonin treatment improved dermatitis severity in DNCB-treated mice accompanied by decreased levels of pro-inflammatory cytokines and chemokines, which probably affected Th2 and Th17 responses and reduced epidermal hyperplasia and lymphocyte infiltration in the skin. Immunomodulary effects were also observed in human keratinocyte HaCaT cells.

The experiments appear to be thoroughly performed and technically sound, and the entire study scientifically accurate. The results are well described in the manuscript and the data interpretation is without overstatements.

While the effects of melatonin are very well described, my main criticism is that there is no deeper insight into the mechanisms of the immunoregulatory activity of melatonin.

Although I consider a very much deeper insight into the mechanisms to be beyond the scope of this study, a few aspects could have been addressed. Especially with the HaCaT cell line, such analyses can be performed relatively easily.

For melatonin signaling two G-protein coupled receptors (MT1 and MT2) are known. However, also retinoid-related and other orphan nuclear receptors have been suggested to mediate some melatonin activity. Authors should address, whether MT1/MT2 is expresses by HaCaT cells. It would be also interesting to know the expression of melatonin receptors in immune cells.

Furthermore, the authors could consider to block receptor function pharmacologically or by knocking out the respective genes.

Beyond that, there are only a few other comments and minor recommendations that should be considered.

1) In figure 1b a clinical score is given determined by summing up a separate scoring of erythema, edema, erosion, and dryness (as described in ref. 20). I think to show the entire clinical picture, authors should present the results of the various clinical symptoms in individual diagrams (in addition to the summary).

2) In a previous study, the authors have used oral melatonin treatment in children with AD and found improved AD disease severity, while scratching activity during sleep was not significantly improved. Proinflammatory cytokines including Th2 cytokines, such as IL-4 and IL-13 (and others such as TSLP, IL-31, IL-33) impact on atopic itch (ref. 22).

Had the authors the chance to investigate here the effect of the topically applied melatonin on itch by addressing the scratching activity of DNCB-treated mice? According to the Materials and Methods section (2.1.5) this has been addressed and thus results should be presented.

3) A 10% melatonin solution dissolved in acetone-olive oil was used for treatment of DNCB-stimulated mice. A 10% melatonin solution is equivalent to approximately 500 mM, while a melatonin solution above 2 mM revealed reduced cell viability of HaCaT cells. First, authors should indicate in the respective Materials and Methods section which solvent was used for the dilution of the melatonin in the in vitro assays. Second, the authors may want to add a short statement why the 10% melatonin solution in acetone-olive oil showed no deleterious effects on skin keratinocytes in vivo.

Minor points:

  • Page 6, line 192: “decrease” should be “decreased
  • Page 8, line 241: “bindd” should be “bind”.
